# Uncovering the hierarchical structure of self-reported hostility

**Martijn W. van Teffelen** *, **Jill Lobbestael, Marisol J. Voncken, Frenk Peeters**

Department of Clinical Psychological Science, Faculty of Psychology and Neuroscience, Maastricht University, Maastricht, Limburg, The Netherlands

* martijn.vanteffelen@maastrichtuniversity.nl

**Data Availability Statement:** Data are available as Supporting Information Files S1_Data.sav and S1_Syntax.sps.

**Funding:** The author(s) received no specific funding for this work.

## Abstract

Hostility and other related terms like anger and aggression are often used interchangeably to describe antagonistic affect, cognition, and behavior. Psychometric studies suggest that hostility consists of multiple separate factors, but consensus is currently lacking. In the present study we examined the hierarchical structure of hostility. The hierarchical structure of hostility was examined in $N = 376$ people (i.e., a mixed community and highly hostile sample), using both specific and broad hostility self-report measures. A series of Principal Components Analyses revealed the structure of hostility at five levels of specificity. At intermediate levels, hostility can consistently be expressed in affective, cognitive, and behavioral components. At the most specific level, hostility can be expressed in terms of Angry Affect; Hostile Intent; and Verbal, Relational, and Physical Aggression. The pattern of associations showed significant convergence, and some divergence with broad and more specific hostility measures. The present findings stress the need for novel instruments that capture each hostility facet separately to reduce conceptual confounding.

## Introduction

In psychological research, human antagonistic behavior and its' related cognitive-affective experiences are often operationalized by the terms anger, hostility, or aggression. Unfortunately, these and other related terms (e.g., irritability, agitation, and frustration) are often used interchangeably. Some use the same term for different constructs (i.e., the jingle fallacy), while others use different terms for the same construct (i.e., the jangle fallacy) [1]. For example, one may refer to hostility as the 'cognitive component' [2, 3], while others refer to hostility as the interrelated elements of cynical beliefs, angry feelings and aggressive responding [4]. Some have referred to anger as the 'affective component' [5, 6], while others refer to anger as a combination of cognitive (i.e., biased information processing) and affective factors [7]. There are researchers who refer to anger as 'irritability' [8]. Also, the behavioral component of hostility is often restricted to observable aggressive behavior [1], while others refer to aggression as the sum of physical/verbal aggressive behavior, attitudinal hostile beliefs and angry responsiveness [9]. Moreover, many self-report measures in the field also use a wide array of terms that even

**Competing interests:** The authors have declared that no competing interests exist.

combine two concepts such as 'anger expression', 'hostile aggression', 'affective aggression' and 'angry hostility', adding further confusion.

Adding to this confusion, two theoretical perspectives can be distinguished that conceptualize hostility and its related cognitive-affective experiences. The first, a *unidimensional perspective* conceptualizes hostility as one construct that includes the interrelated elements of cynical beliefs about others and the world, hostile attribution bias (i.e., the tendency to interpret emotionally ambiguous scenario's as hostile), angry emotional states, and aggressive behavior [4, 10, 11]. The second, a *multidimensional perspective* conceptualizes hostility in terms of a broad conceptual domain that consists out of two or more lower-level facets [2, 3, 12]. Empirical evidence tends to converge with this multidimensional perspective of hostility. That is, exploratory factor analytic (EFA) studies generally find multifactorial solutions [9, 13–15]. Typically, these studies include multiple measurements of hostility and apply EFA to identify the optimal number of factors. Part of the confusion surrounding the concept of hostility can be attributed to diverging results from these exploratory studies. Some previous studies demonstrated two factors such as anger expression and anger experience [14–17]. Others reported three-factor solutions, distinguishing affect, behavior and cognition [14, 18, 19]–also referred to as the ABC-model [20], or similarly the AHA-model (i.e., anger, hostility, aggression) [3]. Finally, also a four factor model has been reported with distinctions between hostility, anger, verbal aggression, and physical aggression [9, 13]. Taken together, the findings of methodologically heterogeneous factor analytic studies hardly converge in terms of number of produced factors and factor content.

A major caveat in the available evidence is that previous work focused on two levels of analysis: a higher order domain, trait, or latent construct (e.g., hostility) and lower-level facets (e.g., experience and expression). Consequently, the outcomes of EFA's are likely to be a function of the combination of instruments, subscales and items that were fed into the respective models. Theoretically the ABC- or AHA-model has been influential. However, empirical evidence shows that the optimal factor structure of hostility is debatable. Moreover, it is unclear how different homogenous facets relate to each other and how central they are to the broad-hostility domain. Lack of consensus leads to measurement imprecision. Close inspection of item-content in widely adapted measures of hostility facets for example shows that items often cross-capture hostility facets. For example, how often one shows certain aggressive behaviors *when angry* (Reactive Proactive Questionnaire) [21], or "I have become so mad that I have broken things" (Buss-Perry Aggression Questionnaire) [9], or "When I get mad I say nasty things" (State-Trait Anger Expression Inventory-2) [6]. Studies in the broad personality psychology field suggest that there is value in investigating model solutions that include more than two conceptual, hierarchical layers [22]. For example, within the construct of narcissism it has been shown that seemingly diverging results of factor analytic studies (i.e., showing different 'optimal' factor solutions) converge into a five-layered hierarchical model in which lower-order facets become more and more specific with each hierarchical layer [23]. Other examples of presumed diverging models that converge into a multi-layered hierarchical model have been reported for agreeableness [24], impulsivity [25], emotion expression [26], and avoidance behavior [27]. Along the same lines, hostility could potentially be expressed as a hierarchical structure consisting of one higher order domain that clusters into two to many facets that become more specific in each additional hierarchical layer. To the best of our knowledge, no hierarchical cluster analysis on hostility has been previously performed.

In sum, factor-analytic evidence tends to converge with a multidimensional view of the hostility construct, but previous work shows differences in number and content of factor solutions. The current study, including facet-level and broad-domain measures, therefore builds

on earlier work by examining the hierarchical structure of the hostility concept. The main expectation is that a multidimensional hierarchical structure will be uncovered.

## Materials and methods

### Participants

Participants were sampled in two ways. First, participants were recruited from the general population in Maastricht, the Netherlands through advertisement. Second, ensuring a representative distribution with enough variation at the extreme end of the hostility dimension (i.e., an estimated 12.4% of the Dutch population show signs of clinically relevant hostility, given that in the Netherlands an estimated 24.5% of people suffer from a mental disorder (i.e., anxiety, mood, eating, personality and somatoform disorders) in one year [28]. Of these individuals an estimated 51% report moderate levels of anger [29]), we actively recruited participants with increased and clinically relevant levels of hostility from two mental health facilities in the Maastricht area (i.e., METggz and U-Center). Patients with a score above 1.22 on the hostility subscale of the Personality Inventory for DSM-5 (PID-5H) were eligible to enter the study. This cutoff equals 1 SD above the observed mean in a Danish population (a comparable population to the Netherlands) and approximates the mean in a clinical population [30]. Exclusion criteria were age younger than 18 and higher than 60, and illiteracy. Patients were excluded from participation by clinical judgement in the mental health facility if they showed signs of current psychosis or mania, alcohol or drug abuse/dependency, and acute suicide risk. For EFA, a minimum sample size of $N = 300$ is suggested [31]. In total, we recruited $n = 347$ people from the general population and $n = 30$ patients with clinically relevant levels of hostility. One patient withdrew consent from the study, so the final sample consisted of $N = 376$. Sample characteristics are shown in Table 1. Statistical analyses showed that, compared to non-patients, patients were less often female, lower educated, student, and were more often using active psychotropic medication.

### Materials

**State trait anger expression inventory-2.**   In the 10-item trait anger scale of the State Trait Anger Expression Inventory-2 (STAXI-2T) [6] items (e.g., "I am hot-headed") are scored on a 4-point Likert scale ranging from one (not at all) to four (very much). Internal consistency, test-retest reliability (e.g., $\alpha = .72 - .96$) and concurrent validity are good and adequate construct validity has been demonstrated [6, 32].

**Aggression questionnaire.**   In the 10-item hostility scale of the Aggression Questionnaire (AQH) [9] items (e.g., "Other people always seem to get the breaks") are scored on a 5-point Likert scale ranging from one (extremely uncharacteristic of me) to five (extremely characteristic of me). Internal consistency (e.g., $\alpha = .73 - .81$) and test-retest reliability are good and adequate/good construct validity has been demonstrated [33, 34].

**Forms of aggression questionnaire.**   The 40-item Forms of Aggression questionnaire (FOA) [35] comprises a list of harmful behaviors measured across five subscales including physical (e.g., "I hit, kick, or push them"), verbal (e.g., "I say mean things to them",) property (e.g., "I damage their property"), relational (e.g., "I ruin their friendships with other people") and passive-rational (e.g., "I criticize their work, even if it is good") aggression. In the original version people are asked to indicate how often each behavior occurs when angry. To minimalize overlap with affective features of hostility, participants in the current study were asked to indicate how often each behavior occurs *in general* instead. Items are scored on a 5-point Likert scale ranging from one ((almost) never) to five ((almost) always). Good internal

**Table 1. Sample characteristics.**

| | Total sample (N = 376) | Stratum Non-clinical (n = 347) | Clinical (n = 29) | Statistical dif. $ZZ / X^2 / t$ (p) |
|---|---|---|---|---|
| Age in years, mean (SD) | 35.15 (14.72) | 34.93 (14.97) | 37.72 (11) | 1.47 (.141) |
| Female, n (%) | 280 (74) | 266 (77) | 14 (48) | 11.34 (.001) |
| Male, n (%) | 96 (26) | 81 (23) | 15 (52) | |
| Nationality, n (%) | | | | 2.05 (.563) |
| Dutch | 353 (95) | 324 (93) | 29 (100) | |
| Belgian | 13 (3) | 13 (4) | 0 (0) | |
| German | 4 (1) | 4 (1) | 0 (0) | |
| Other | 6 (2) | 6 (2) | 0 (0) | |
| Education, n (%) | | | | 27.46 (< .001) |
| Low | 91 (24) | 89 (26) | 2 (7) | |
| Middle | 141 (38) | 114 (33) | 24 (83) | |
| High | 144 (38) | 141 (41) | 3 (10) | |
| Work situation, n (%) | | | | 26.14 (< .001) |
| Employed | 155 (41) | 141 (41) | 14 (48) | |
| Unemployed | 42 (11) | 36 (10) | 6 (21) | |
| Student | 135 (36) | 135 (39) | 0 (0) | |
| Social security | 41 (11) | 32 (9) | 3 (10) | |
| Retired | 3 (1) | 3 (2) | 0 (0) | |
| Medication use, n (%) | | | | |
| Antidepressant, SSRI | 24 (6) | 16 (5) | 8 (28) | 23.64 (< .001) |
| Antidepressant, SNRI | 10 (3) | 5 (1) | 5 (17) | 25.81 (< .001) |
| Antidepressant, TCA | 2 (1) | 2 (1) | 0 (0) | .17 (.682) |
| Antidepressant, other | 3 (1) | 2 (1) | 1 (3) | 2.79 (.095) |
| Antipsychotic, atypical | 6 (2) | 4 (1) | 2 (7) | 5.62 (.018) |
| Anxiolytic | 10 (3) | 5 (1) | 5 (17) | 25.81 (< .001) |
| Mood stabilizer | 3 (1) | 2 (1) | 1 (3) | 2.79 (.095) |
| Stimulant | 7 (2) | 6 (2) | 1 (3) | .43 (.511) |
| Study variables | | | | |
| STAXI-2T, mean (SD) | 16.67 (5.21) | 16.01 (4.53) | 24.76 (6.03) | -7.63 (< .001) |
| AQH, mean (SD) | 19.01 (7.25) | 18.47 (7.02) | 25.45 (7.04) | -5.13 (< .001) |
| FOA, mean (SD) | 56.72 (13.12) | 55.57 (11.99) | 70.52 (17.78) | 4.89 (< .001) |
| PID-5H, mean (SD) | .74 (.59) | .66 (.51) | 1.65 (.68) | -7.69 (< .001) |

SSRI = selective serotonergic reuptake inhibitor; SNRI = selective noradrenergic reuptake inhibitor; TCA = tricyclic antidepressant.

consistency (e.g., α = .93 - .94) and adequate construct, convergent and discriminant validity have been demonstrated [35].

**Personality inventory for dsm-5.** In the 10-item hostility scale of the Personality Inventory for DSM-5 (PID-5H) [36] items (e.g., "I snap at people when they do little things that irritate me") are scored on a 4-point Likert scale ranging from zero (very false or often false) to three (very true or often true). Internal consistency (e.g., α = .88 - .90) is good and adequate construct and convergent validity have been demonstrated [30, 37–39].

## Procedure

The Ethical Review Committee Psychology and Neuroscience at Maastricht University provided ethical approval to carry out the study (ERCPN- 167_08_05_2016). The study was pre-

registered at https://osf.io/gpju6. Some protocol changes were made after the study was preregistered. Specifically, we decided to perform a different analytical approach, we chose to study hostility at trait level (instead of at both state and trait levels) and patients were not screened for instable use of psychotropic medication. The study was performed completely online using Qualtrics software. Beforehand, people were told that the study was about investigating the relationship between thoughts, feelings, and behaviors. After participants provided informed consent, an online link to the study's questionnaires was sent by e-mail. Then, information about demographic variables, use of psychotropic medication were obtained and the PID-5H, STAXI-2T, AQH, and FOA were administered. After completion, participants were debriefed and received course credit or participated in a raffle with 347 times €7,50 worth of rewards.

## Statistical analysis

IBM SPSS and Amos version 24 were used for statistical analysis. First, descriptive statistics were calculated. Second, Spearman's correlation analysis was run to examine baseline correlations between all multidimensional and unidimensional hostility measures. Then, the hierarchical structure of hostility was examined using the so-called 'Bass-Ackwards' method [40]. Within the context of the examination of personality models, Goldberg [40] provided a method of examining hierarchical structures in models with more than two levels. The approach allows for the examination of various hierarchical levels of specificity, from a broad construct to more fine-grained, lower-level facets that become more specific at each hierarchical level. Factor solutions were identified using Principal Components Analysis (PCA). First, one unrotated principal component was extracted, followed by the extraction of successively (i.e., two, three, etc.) more Varimax rotated principal components. Varimax rotation was opted following Goldberg [40] because of optimal parsimony and to encourage factor markers that are maximally unrelated to each other. This was then repeated until one of the factors was either too specific to be interpreted (e.g., containing one item) or was no longer interpretable (e.g., by containing items that show hardly any content similarity). After each extraction, factor loadings were saved and correlated to compare relationships at each level. The identified principal components were then correlated with the raw scores of the questionnaires. The minimal anonymized data set and accompanying syntax are presented in S1 Syntax and S1 Data.

## Results

Means and standard deviations of study variables are shown in Table 1. All study variables were positively skewed (i.e., the value 0 is outside the +/- 2 * standard error interval of the skewness value). All scores resembled those of other studies using population samples [32, 34, 35, 41–43]. Univariate outlier inspection of study variables revealed no bimodality or consistent univariate outliers (following the 3 * interquartile range criterion), suggesting that the patients in the present sample did not form a data-cluster.

First, Spearman's correlations and Cronbach's $\alpha$'s are shown in Table 2. These results suggest that the STAXI-2T, AQH, FOA and PID-5H are significantly positively interrelated. All correlations were large according to Cohen [44] except for the relationship between the FOA and AQH, that was medium.

Then, to examine the hierarchical structure of hostility a PCA was run using the 'Bass-Ackwards' method. We evaluated multivariate normality and linearity by inspecting Mahalanobis distance. We observed two multivariate outliers who were removed from the analysis. We observed non-normality on eighteen FOA items (i.e., skewness values smaller or larger than three standard errors) [45]. Of these eighteen items fifteen extremely violated the normality assumption even after inverse-transformation (i.e., $1/x$) and were removed from further

**Table 2. Spearman's rho correlations between uni- and multidimensional hostility constructs.**

|  | STAXI-2T | AQH | FOA | $\alpha$ |
|---|---|---|---|---|
| STAXI-2T |  |  |  | .88 |
| AQH | .54* |  |  | .86 |
| FOA | .60* | .43* |  | .93 |
| PID-5H | .79* | .51* | .65* | .88 |

*Note.*

*$p < .001$.

Cronbach's $\alpha$ is reported in the diagonal.

analyses to maintain model robustness. The removed items are shown in S1 Table. Factor loadings are presented in S2 Table. S1 Fig shows the decision process for principal component extraction. First, one unrotated principal component was extracted, followed by the extraction of successively (i.e., two, three, etc.) more Varimax rotated principal components. This was then repeated until one of the factors was either too specific to be interpreted (e.g., containing one item) or was no longer interpretable (e.g., by containing items that show hardly any content similarity). The first unrotated principal component accounted for 30% of the total variance. The first ten eigenvalues were: 16.16, 3.90, 2.68, 2.12, 1.61, 1.35, 1.24, 1.20, 1.13, and 1.06. Then, successively larger solutions (i.e., two, three, etc.) were examined. Inspection of the 6-principal component solution showed that the last factor consisted of the two items: "I resent being told what to do, even by people in charge" and "I feel annoyed when not given recognition for doing good work". Thus, the 6-principal component solution was interpreted as not meaningful, resulting in a 5-principal component solution as base of the hierarchical model, accounting for 49% of the variance.

The hierarchical 5-principal component model is shown in Fig 1. Correlations between the component loadings and the original scales are shown in Table 3. Rotated component loadings and item content are shown in S2 Table. All principal components were labeled according to what was most common to all these items. The first component (P1.1) was labeled Hostility and demonstrated significant positive associations to the original hostility scales ranging from $r = .66$ (AQ-H) to $r = .88$ (PID-5H). The principal components in the two-factor solution were

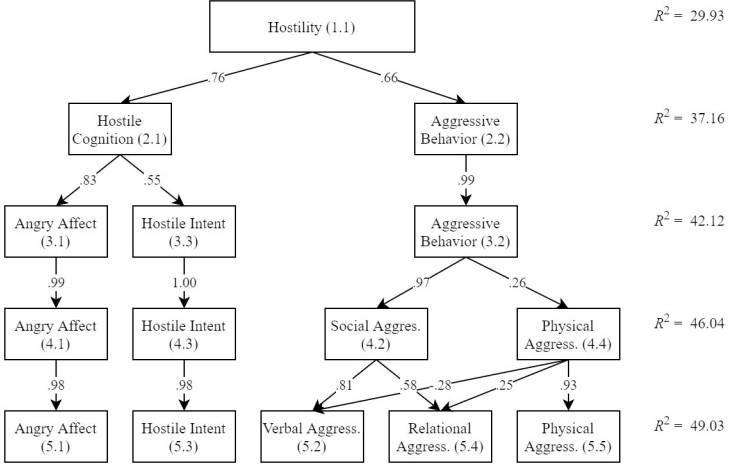

**Fig 1. Hierarchical structure of hostility.**

**Table 3. Spearman's rho correlations between factor scores and hostility instruments.**

|  | STAXI-2T | AQH | FOA | PID-5H |
|---|---|---|---|---|
| P1.1 | .84*** | .66*** | .87*** | .88*** |
| P2.1 | .80*** | .77*** | .33*** | .76*** |
| P2.2 | .35*** | .14** | .92*** | .42*** |
| P3.1 | .74*** | .18*** | .28*** | .75*** |
| P3.2 | .30*** | .18*** | .90*** | .36*** |
| P3.3 | .35*** | .92*** | .20*** | .29*** |
| P4.1 | .75*** | .18*** | .28*** | .76*** |
| P4.2 | .34*** | .18*** | .87*** | .38*** |
| P4.3 | .34*** | .92*** | .19*** | .29*** |
| P4.4 | -.17** | .00 | .11* | -.08 |
| P5.1 | .71*** | .16** | .20*** | .72*** |
| P5.2 | .35*** | .13* | .72*** | .37*** |
| P5.3 | .35*** | .93*** | .20*** | .30*** |
| P5.4 | .02 | .10 | .35*** | .05 |
| P5.5 | -.12* | .04 | .23*** | -.03 |

* significant at $p < .05$;

** significant at $p < .01$;

*** significant at $p < .001$.

$R^2$-values of each level are respectively: 29.93, 37.16, 42.12, 46.04, and 49.03.

labeled Hostile Cognition (P2.1) and Aggressive Behavior (P2.2). Hostile Cognition related most strongly to the total scores of the STAXI-2T, AQH and PID-5H. Aggressive Behavior most strongly related to the FOA ($r = .92$). In the three-component solution the Hostile Cognition component split into an Angry Affect (P3.1) and Hostile Intent (P3.3) component. Angry Affect most strongly related to the STAXI-2T and PID-5H. Hostile Intent most strongly related to the AQH ($r = .92$). In the four-component solution items from the Aggressive Behavior component split into a Social Aggression (P4.2) and Physical Aggression (P4.4, including many inverse-transformed items) components. Social Aggression related most strongly to the FOA ($r = .87$), whereas Physical Aggression related most strongly to the STAXI-2T ($r = -.17$, $p = .001$) and FOA ($r = .11$, $p = .029$). In the five-component solution content from the Social Aggression component split into a Verbal Aggression (P5.2) and Relational Aggression (P5.4) component. Verbal Aggression ($r = .72$) and Relational Aggression ($r = .35$) both most strongly related to the FOA.

## Discussion

The present study is, to the best of our knowledge, the first to explore the hierarchical structure of self-reported trait-hostility. We predicted that hostility can be defined as a construct that can be interpreted at different levels of specificity or, in other words, that hostility shows a multidimensional hierarchical structure. We observed that at the highest, most abstract level Hostility is characterized by a low threshold to experience and react harmfully upon angry affect. Findings demonstrate large positive associations between hostility and raw scores on different instruments of hostility. This finding is consistent with different conceptualizations of hostility [4, 8, 10, 11, 46, 47]. At the second level, hostility splits up into an experiential (Hostile Cognition) and expressive component (Aggressive Behavior), converging with factor analytic studies [14–17]. Correlations with the original scales show that experiential aspects of

hostility are mostly captured by the STAXI-2T, AQH and PID-5H, whereas the expressive aspects are mostly captured by the FOA. At the third level, the experiential factor splits up into an affective (Angry Affect) and cognitive factor (Hostile Intent). This is in line with factor analytic studies that demonstrated a cognitive, affective and behavioral hostility factor [14, 18, 19]. Correlations with the original scales show that the affective component is mostly captured by the STAXI-2T and PID-5H, whereas the cognitive component is mostly captured by the AQ-H. At the fourth level, the behavioral component differentiated in an interpersonal (Social Aggression) and physical (Physical Aggression) component. This largely converges with studies showing a four-factor solution consisting of a cognitive, affective and two behavioral factors [9, 13]. Associations with the original scales show that Social Aggression is mostly captured by the FOA, whereas Physical Aggression was mostly captured by the STAXI-2T and the FOA. Moreover, we demonstrated that the interpersonal component split up into a verbal and relational aggression component. In short, the present findings show that seemingly diverging factor analytic solutions from previous studies converge into one hierarchically structured model of hostility.

Similar to other models for which hierarchical structures have shown value (e.g., narcissism, agreeableness, impulsivity, avoidance behavior, emotional expression), the current research demonstrates that at the highest, most abstract level 30% of the variance in hostility is explained by one underlying dimension. Already at the second hierarchical level behavioral characteristics are separated from cognitive characteristics, showing that behavior is a clear distinct characteristic within hostility. Moving down another hierarchical layer, interpretational characteristics are separated from affective characteristics. The affective and interpretational components of hostility remain stable facets in the majority (i.e., three out of five) of hierarchical layers, marking their relative stability. At even more specific hierarchical layers, behavioral characteristics of hostility differentiate in three expressive forms of aggressive behavior: physical, verbal, and relational aggression. Together, these five facets explain 49% of the variance in hostility items. Surprisingly, the Physical Aggression component at level four and five showed a negative association with the STAXI-2T. A likely explanation is that all inversely-transformed items are included in the Physical Aggression component, and that the STAXI-2T includes many items that tap into physical aggressive behavior (e.g., "When I get mad I say nasty things"). Overall, these findings show that hostility can be perceived as multifaceted construct in which affective, interpretational, and behavioral characteristics are stable components.

Several limitations impact the present findings. First, the present work did not include any predictive measures. Although the present findings show convergent validity, we cannot draw any definite conclusions on the criterion validity of the present findings. A recommendation for future research is hence to include instruments that show differential relationships to different hostility facets, such as agreeableness, shame proneness, empathy, trust, and compassion. Second, the majority of the sample (74%) was female. Given that women exhibit more indirect forms of aggression (e.g., relational, or passive-rational aggression) and men exhibit more direct forms (e.g., physical aggression) [48], the results might differ from a sample that includes more men. Third, 30 patients were included in the present sample to ensure enough variation at the extreme end of the hostility dimension. Network models of psychopathology suggest that overall symptom severity is positively related to the strength of correlations between symptom clusters [49]. Recent work shows that hostility is associated with increase psychopathological severity [50]. Theoretically it could be possible that different patterns in the patient subgroup may impact the present findings, for example by artificially driving up correlations. Nonetheless, absence of bimodality and univariate outliers suggests that hostility levels in the present sample reflect a distribution that might be expected in the population and is in line with the dimensional approach to psychopathology [51]. Fourth, all self-reports were

administered in Dutch. Consequently, the results of present work may be culturally bias and may not generalize to non-Dutch cultures. Fifth, the present work approached hostility from a trait approach. As a result, self-report measures were used. A recommendation for future research is to include measures that capture (state) aspects of hostility on different analytical levels, such as physiologically (e.g., variations in heart-rate variability or skin conductance) or behaviorally (e.g., Competitive Reaction Time Task or Point Subtraction Aggression Paradigm [52, 53]. Last, in the present study we worked with a selection of instruments that measure hostility constructs. The STAXI-2 and AQ show excellent psychometric properties [6, 32–34] and are extensively used and cited—A Web of Science citation report on 15 November 2019 reveals that the STAXI(-2) and AQ are cited respectively 6160 and 4139 times in scientific articles since 1988. The FOA is less commonly used and cited, but does show good psychometric properties in its original form and closely fits modern definitions of aggressive behavior (i.e., *any* behavior that is intended to cause harm to another person) [1]. Despite this careful questionnaire selection, future research needs to examine whether this hierarchical structure holds when more or other instruments are used.

The current finding that hostility is a construct that can be interpreted at different levels of theoretical generality versus specificity comes with several main implications. One implication is that a person can score high on a measure that captures one aspect of hostility but will not necessarily score high on another. For example, a person with a tendency to be easily angered does not necessarily easily engage in aggressive behavior. Also, a person with the tendency to respond physically aggressive, does not necessarily have the tendency to be verbally aggressive. Hostile affect, cognition and behavior may therefore have different antecedents and consequences, requiring a different approach in clinical context. More importantly, the current study illustrates that the lack of consensus in the current hostility literature is likely the result of conceptual identity confusion (i.e., jingle and jangle fallacies). This, in turn, leads to reduced measurement precision and a fragmentation of the research field (i.e., two researchers could study the same construct but name their construct differently). Note that hostility is investigated not only in the field of aggression, but also the fields of social psychology, clinical psychology, and psychiatry. This underlines the need to be both more critical towards the use of language and to be very precise in choice of measurement instruments in these fields. This includes, for instance, the use of items that cross-capture hostility facets, while they pretend to measure only one facet (see S2 Table for examples of items that capture multiple facets). Hopefully, this study will stimulate joint efforts to move towards the standardized use of hostility and its subcomponents. Moreover, we hope to contribute towards moving the field of aggression research to a more valid and standardized assessment level by further stimulating and ameliorate the accurate and standardized assessment and operationalization of hostility.

## Supporting information

**S1 Data. Minimal anonymized data set.**
(SAV)

**S1 Syntax. Syntax accompanying the minimal anonymized data set.**
(SPS)

**S1 Table. FOA items that were removed due to extreme normality violation.**
(DOCX)

**S2 Table. Item content and loading for all rotated principal component solutions.**
(XLSX)

**S1 Fig. Decision flow chart Bass-Ackwards approach.**
(TIF)

## Acknowledgments

There are no acknowledgments for the present work.

## Author Contributions

**Conceptualization:** Martijn W. van Teffelen, Jill Lobbestael, Marisol J. Voncken, Frenk Peeters.

**Formal analysis:** Martijn W. van Teffelen.

**Investigation:** Martijn W. van Teffelen.

**Methodology:** Martijn W. van Teffelen, Jill Lobbestael, Marisol J. Voncken, Frenk Peeters.

**Project administration:** Martijn W. van Teffelen.

**Supervision:** Jill Lobbestael, Frenk Peeters.

**Validation:** Jill Lobbestael, Marisol J. Voncken, Frenk Peeters.

**Writing – original draft:** Martijn W. van Teffelen, Frenk Peeters.

**Writing – review & editing:** Jill Lobbestael, Marisol J. Voncken.

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
