## [Decision Letter · Decision Letter 0]

13 Aug 2020

PONE-D-20-17138

Uncovering the hierarchical structure of self-reported hostility

PLOS ONE

Dear Dr. van Teffelen,

Thank you for submitting your manuscript to PLOS ONE. After careful consideration, we feel that it has merit but does not fully meet PLOS ONE’s publication criteria as it currently stands. Therefore, we invite you to submit a revised version of the manuscript that addresses the points raised during the review process.

As the reviewers point out there is a need to clarify the gap in hostility research more clearly and a need to justify statistical decisions and sampling in your reserach.

We look forward to receiving your revised manuscript.

Kind regards,

Svenja Taubner

Academic Editor

PLOS ONE

Journal Requirements:

Reviewers' comments:

Reviewer's Responses to Questions

**Comments to the Author**

1. Is the manuscript technically sound, and do the data support the conclusions?

Reviewer #1: Partly

Reviewer #2: Partly

2. Has the statistical analysis been performed appropriately and rigorously? 

Reviewer #1: Yes

Reviewer #2: Yes

3. Have the authors made all data underlying the findings in their manuscript fully available?

Reviewer #1: No

Reviewer #2: No

4. Is the manuscript presented in an intelligible fashion and written in standard English?

Reviewer #1: Yes

Reviewer #2: Yes

5. Review Comments to the Author

Reviewer #1: The article by van Teffelen and colleagues is clear in its presentation, sound in its statistical approach and addresses an interesting topic in conceptual hostility research. Nevertheless, more clarification on statistical decisions is needed in my opinion. First, the authors should address, either in the participants section or at least in the limitations, how the 30 patients might have influenced the results of the analysis. Networkmodels of psychopathology suggest that the strength of the correlation between different symptoms might be directly related to the overall symptom severity of an individual. Consequently, since PCA cannot control for different subgroups (likewise EFA at n = 30) the correlations between the rotated factors and the raw scores might be influenced by different patterns in both groups. While it seems like the logic behind including these 30 patients was to increase the variance of hostile traits in the sample, I could not find a clear statement on why the authors found it necessary to include this subgroup. Second, the correlations of the 4th and 5th principal components with the raw scores seem to require more in depth discussion. For example, the component dubbed “Physical Aggression” P4.4 is most strongly related to the inverse of the STAXI-2T score. While this might not be unexpected for researchers more familiar with these measures, for me it begets the question how a measure assessing anger is inversely correlated with supposed physical aggression. If this is obvious, please forgive me. Additionally, components like P5.5 seem to have overall low (albeit significant) associations with all scores which makes me wonder if the gain in explained variance is justified. An addition to table 3 showing the increase in R² at each step would be helpful. Thank you for your work and excuse my comments if inappropriate.

Reviewer #2: First of all, I very much appreciate the effort of the authors to show some light on a construct like hostility where there seems to be no clear consensus yet. The AHA model (aggression, hostility, and anger) is the most accepted model and the one that is most clearly understood by assigning each part of the behavior to a specific dimension. Thus, anger is emotional, aggression is the behavioral and hostility is cognitive ot the respectively dimensions of the AHA. Therefore, I do not finish seeing in what this work contributes to show a hierarchical structure on hostility as a whole instead of as a part of the AHA.

I would like to see clearly what is the gap of a specific literature review that this work tries to fill or cover. They did not convince me yet.

To compare the tradition of psychometric studies of personality and intelligence with hostility is not pretty accurate and it is not adequate to include it as a premise to conduct this study. It is just an opinion and not a suggestion

Results should provide more tables and flow decisions according to APA style for psychometric studies. For example, I needed to read some psychometric-key elements in the MS:

Accuracy: The closeness of a measurement, calculation, or specification to the correct value. Contrasted with precision (the degree of refinement of the measurement, etc.)

Appropriate: Suitable or proper in the circumstances

Validity: Well-founded on fact, or established on sound principles, and thoroughly applicable to the case or circumstances; soundness and strength (of argument, proof, authority, etc.)

The study was accurate but I found some lack of appropriate (cultural biases in the sample, clinic and non-clinic participants) and validity (especially, predictive validity, supplemental documentation with a final scale using or suggesting items, divergent instruments, etc..)

I need that authors demonstrate why this study is really necessary and what gaps they tried to fill in the measurement of the hostility.

6. PLOS authors have the option to publish the peer review history of their article (what does this mean?). If published, this will include your full peer review and any attached files.

Reviewer #1: No

Reviewer #2: **Yes: **Jose M Mestre ORCID: https://orcid.org/0000-0002-6822-4970

---

## [Decision Letter · Decision Letter 1]

10 Sep 2020

Uncovering the hierarchical structure of self-reported hostility

PONE-D-20-17138R1

Dear Dr. Willem van Teffelen,

We’re pleased to inform you that your manuscript has been judged scientifically suitable for publication and will be formally accepted for publication once it meets all outstanding technical requirements.

Your revision was adressing all issues raised by the reviewers and both the reviewer and me are impressed with your efforts.

Kind regards,

Svenja Taubner

Academic Editor

PLOS ONE

Additional Editor Comments (optional):

Reviewers' comments:

Reviewer's Responses to Questions

**Comments to the Author**

1. If the authors have adequately addressed your comments raised in a previous round of review and you feel that this manuscript is now acceptable for publication, you may indicate that here to bypass the “Comments to the Author” section, enter your conflict of interest statement in the “Confidential to Editor” section, and submit your "Accept" recommendation.

Reviewer #2: (No Response)

2. Is the manuscript technically sound, and do the data support the conclusions?

Reviewer #2: (No Response)

3. Has the statistical analysis been performed appropriately and rigorously? 

Reviewer #2: (No Response)

4. Have the authors made all data underlying the findings in their manuscript fully available?

Reviewer #2: (No Response)

5. Is the manuscript presented in an intelligible fashion and written in standard English?

Reviewer #2: Yes

6. Review Comments to the Author

Reviewer #2: Honestly, I am impressed by their comments and efforts to carry out an improved paper. They have addressed every single concern that I had

7. PLOS authors have the option to publish the peer review history of their article (what does this mean?). If published, this will include your full peer review and any attached files.

Reviewer #2: **Yes: **Jose M Mestre, Department of Psychology, Universidad de Cádiz, Spain

---

## [Editor Report · Acceptance letter]

15 Sep 2020

PONE-D-20-17138R1

Uncovering the hierarchical structure of self-reported hostility

Dear Dr. van Teffelen:

I'm pleased to inform you that your manuscript has been deemed suitable for publication in PLOS ONE. Congratulations! Your manuscript is now with our production department.

Kind regards,

on behalf of

Dr. Svenja Taubner 

Academic Editor

PLOS ONE